# CtIP forms a tetrameric dumbbell-shaped particle which bridges complex DNA end structures for double-strand break repair

Oliver J Wilkinson[1], Alejandro Martín-González[2], Haejoo Kang[3], Sarah J Northall[1], Dale B Wigley[3], Fernando Moreno-Herrero[2], Mark Simon Dillingham[1]*

[1]School of Biochemistry, University of Bristol, Bristol, United Kingdom; [2]Department of Macromolecular Structures, Centro Nacional de Biotecnologia, Consejo Superior de Investigaciones Cientificas, Madrid, Spain; [3]Department of Medicine, Imperial College London, London, United Kingdom

**Abstract** CtIP is involved in the resection of broken DNA during the S and G2 phases of the cell cycle for repair by recombination. Acting with the MRN complex, it plays a particularly important role in handling complex DNA end structures by localised nucleolytic processing of DNA termini in preparation for longer range resection. Here we show that human CtIP is a tetrameric protein adopting a dumbbell architecture in which DNA binding domains are connected by long coiled-coils. The protein complex binds two short DNA duplexes with high affinity and bridges DNA molecules in trans. DNA binding is potentiated by dephosphorylation and is not specific for DNA end structures per se. However, the affinity for linear DNA molecules is increased if the DNA terminates with complex structures including forked ssDNA overhangs and nucleoprotein conjugates. This work provides a biochemical and structural basis for the function of CtIP at complex DNA breaks.
DOI: https://doi.org/10.7554/eLife.42129.001

*For correspondence:
mark.dillingham@bristol.ac.uk

**Competing interests:** The authors declare that no competing interests exist.

## Introduction

DNA double-strand breaks (DSBs) are a potentially lethal form of DNA damage associated with genomic instability, gross chromosomal rearrangements and apoptosis (*Ranjha et al., 2018*). They are caused by exogenous agents such as ionising radiation (IR) and chemotherapeutic drugs, but also by normal cell metabolism such as during V(D)J/class-switch recombination, Spo11-mediated meiotic recombination, or replication fork collapse. DSBs are repaired by two major pathways: homologous recombination (HR) and non-homologous end-joining (NHEJ). The predominance of either pathway is regulated in a cell cycle-dependent manner owing to the requirement for a sister chromatid to act as a template for error-free repair. HR is promoted in S and G2 phases of the cell cycle by a variety of mechanisms including activation of CtIP (*Huertas and Jackson, 2009*; *Shibata and Jeggo, 2014*).

The process of DNA end resection is a critical regulatory step, which commits a cell to repair by HR. During end resection, DSBs are preferentially degraded by nucleases to reveal long 3′-ssDNA overhangs (*Cejka, 2015*; *Daley et al., 2015*). These are stabilised by RPA and then bound by Rad51, which forms nucleoprotein filaments that undergo strand invasion with the sister chromatid, eventually leading to DNA repair without loss of genetic information (*Kowalczykowski, 2015*). DNA end resection can be broadly separated into three mechanistic stages: initial recognition of the DSB, short-range resection and long-range resection. Recognition of the DSB is achieved by binding of the Mre11-Rad50-Nbs1 complex (MRN) (*Paull and Gellert, 1999*) which activates the DNA damage response through its interaction with ATM, and subsequently recruits other repair factors to the

damage site including CtIP, which is itself activated by cyclin-dependent kinase in an Mre11-mediated process (*Buis et al., 2012*; *Huertas and Jackson, 2009*). Human CtIP, which is mutated in Seckel2 and Jawad syndromes (*Qvist et al., 2011*), has been shown to be crucial for the promotion of end resection and HR-mediated repair in many genetic studies (*Huertas and Jackson, 2009*; *Sartori et al., 2007*; *You et al., 2009*), but exactly how MRN and CtIP co-operate to achieve short-range resection is not fully understood. MRN harbours both 5′−3′ endonuclease and 3′−5′ exonuclease activities in the Mre11 subunit, and these activities are mediated by CtIP (*Cannavo and Cejka, 2014*; *Paull and Gellert, 1999*). Some reports suggest that CtIP and its orthologues possess intrinsic endonuclease activity (*Lengsfeld et al., 2007*; *Makharashvili et al., 2014*; *Wang et al., 2014*), but the primary structure of the protein is not obviously related to any known class of nuclease, and other groups have reported that their preparations are devoid of nuclease activity (*Andres and Williams, 2017*). Whatever the case, the combined activities of MRN and CtIP are especially important for processing complex DNA break structures, where the ends are composed of damaged nucleotides, non-canonical DNA secondary structures, or covalent nucleoprotein complexes (*Andres and Williams, 2017*; *Aparicio et al., 2016*; *Aparicio and Gautier, 2016*; *Hartsuiker et al., 2009a*; *Hartsuiker et al., 2009b*; *Paudyal et al., 2017*; *Quennet et al., 2011*). Short range nucleolytic processing of these ends by MRN-CtIP facilitates further long-range resection by processive helicases and nucleases including Exo1, DNA2, BLM, and WRN (*Cejka, 2015*; *Daley et al., 2017*).

In vitro experiments with recombinant human proteins show that Mre11 can cut the 5′-strand of a DNA duplex downstream of a break. This activity is dependent on the presence of Nbs1 and enhanced by a nucleoprotein block at the 5′-terminus (*Anand et al., 2016*; *Reginato et al., 2017*; *Wang et al., 2017*). In Xenopus egg cell-free extracts, it was shown that CtIP-MRN is essential for the removal of Top2-DNA adducts and the subsequent resection of these breaks (*Aparicio et al., 2016*). Furthermore, the interaction between CtIP and BRCA1 (mediated by phosphorylation of S327 on CtIP) is required for the processing of Top2-adducted 'complex' DSBs but dispensable for the resection of endonuclease-generated 'simple' breaks (*Aparicio et al., 2016*). This is reminiscent of the finding that the CtIP mutant N289A/H290A cannot rescue the end resection-deficient phenotype for topoisomerase-induced breaks but is proficient for simple breaks, although here it was postulated that this deficiency was based on a disruption of CtIP endonuclease activity (*Makharashvili et al., 2014*). In any case, once a nick is made, Mre11 uses its 3′−5′ exonuclease function to strip back the 5′ strand towards the blocked end. In these experiments, wild type CtIP did not enhance the endonuclease activity of MRN, although a stimulation of activity was observed when a phosphomimic mutant (T847E/T859E) was used (*Deshpande et al., 2016*). In mammalian cells, it was shown that both phosphorylation sites are important for CtIP function. In contrast, studies using recombinant *S.cerevisiae* proteins show that the CtIP orthologue Sae2 appears to be absolutely required for unlocking a cryptic endonuclease activity of Mre11, possibly highlighting a difference in regulation between the yeast and human systems (*Cannavo and Cejka, 2014*).

Structural information for CtIP is limited to a small N-terminal region of the *S.pombe* homologue Ctp1 (amino acids 5–60) and a similar region of human CtIP (amino acids 18–52) (*Andres et al., 2015*; *Davies et al., 2015*). In both cases, there is an interlocking arrangement of two antiparallel coiled coils which leads to a 'dimer of dimers' arrangement. In addition, it has been shown that a truncated form of the human protein comprising residues 18–145 forms a stable homotetramer in vitro (*Davies et al., 2015*). A mutation (L27E) that prevented tetramerization also rendered CtIP non-functional in vivo. The proposed model that arises from these reports is one where two rigid coiled coils (aa1-145) protrude in opposite directions from the tetramerisation domain, spanning a distance of about 30 nm, with the unstructured C-termini presumably placed at the opposing ends (*Forment et al., 2015*). It has been shown that CtIP binds to DNA; for S. *pombe* Ctp1 a number of DNA substrates have been interrogated by EMSA and were all shown to be bound with similar affinities (*Andres et al., 2015*). This study also used scanning alanine mutagenesis to identify residues that are important for binding. A C-terminal 'RHR' motif (equivalent to conserved residues R837 to R839 in the human protein) was shown to be crucial for DNA recognition, but it was also shown that an N-terminal part of the protein binds DNA. For human CtIP, it has been shown that the C-terminal region (aa769-897) interacts with a 200 bp duplex, whereas the N-terminal region (aa1-145) does not (*Davies et al., 2015*). In the case of the full length human protein, DNA binding activity has only been demonstrated using gel-based crosslinking assays which gives limited information about equilibrium binding affinities (*Makharashvili et al., 2014*). An important outstanding question is whether

the affinity of CtIP for different DNA end structures can explain its relative importance in the processing of 'complex' versus 'simple' ends in vivo. Moreover, since the DNA binding domains are thought to reside in the opposing C-termini, this may be a way in which a CtIP tetramer could bridge two distant broken DNA ends to promote DSB repair.

In this report, we show that full length human CtIP is a tetrameric protein that forms a dumbbell-shaped particle consisting of two polar globular domains separated by a thin and flexible 'rod'. Semi-quantitative measurements of CtIP affinity for a range of DNA substrates show that, somewhat surprisingly, CtIP binding does not require DNA ends per se. However, CtIP binds much more effectively to structures with DNA ends that contain model nucleoprotein blocks or single-stranded DNA fork structures. Our results are rationalised in terms of a 'slide and capture' model for the recognition of complex DNA ends. Furthermore, we provide direct evidence for CtIP-dependent DNA bridging activity and show that both DNA binding and bridging are reduced by mutations which target the tetramerization interface or the C-terminal RHR motif.

## Results

### Full length human CtIP is a tetrameric protein that forms a dumbbell-shaped particle

Full length recombinant human CtIP and mutant variants were overexpressed in insect cells and purified to homogeneity using a cleavable C-terminal strep-tag (*Figure 1*; see Materials and methods for details). Analysis by SEC-MALS showed that wild type CtIP ran as a single symmetrical peak with a predicted molecular weight that was equal within error to a tetrameric species (391 ± 23 kDa; *Figure 1A*), as we had expected based on previous work with the isolated N-terminal region of the protein (*Andres et al., 2015*; *Davies et al., 2015*). However, the CtIP peak elutes earlier in the size exclusion column than would be expected for a ~ 400 kDa protein, which suggests that it adopts an unusual shape. Indeed, direct observation of purified CtIP using negative stain electron microscopy revealed an extraordinary dumbbell-like structure in which polar globular domains are held about 30 nm apart by a central 'rod' (*Figure 1B*, β class; *Figure 1—figure supplement 1*). The length of the rod is broadly consistent with the predicted length of the coiled-coil domains, which suggests that the centre of the dumbbell is the site of the tetramerization interface as proposed in a previous model (*Forment et al., 2015*). In addition to the dumbbell structures, we less frequently observed structures that we call 'tadpoles' containing a single globular domain and a tail, which presumably represent a dimeric form of the protein formed during EM sample preparation and deposition (*Figure 1B*, α class; *Figure 1—figure supplement 1*).

Broadly similar structures were observed by atomic force microscopy, although the sample appeared more heterogenous when applied to a mica surface, and five classes of particles were distinguished based on their volume and morphology (*Figure 1C*). The simplest and smallest structures consisted of a single bright spot or a single spot with a tail (class I and II). Somewhat larger structures were observed in which two or three spots are connected by rods (class III and IV respectively). Finally, the largest class of particle contained up to four connected spots (class V). These different particles were associated with different volume profiles, with the largest particle class (class V) displaying a relative volume consistent with a tetrameric species (~380 kDa; *Figure 1D*) (*Fuentes-Perez et al., 2012*). These either resembled the dumbbells we had observed by EM (β class particles) or showed a morphology that suggested that the polar globular domains had separated ('splayed dumbbells'; see *Figure 1—figure supplement 2*). Other particle classes presumably arise from dissociation of the tetramer into smaller species, either upon dilution for the AFM analysis or because of deposition onto the mica surface. To aid interpretation of the different particle classes and for further biochemical analysis (see below) we also analysed two mutant proteins (*Figure 2*). The putative DNA binding mutant CtIP[R839A] (*Andres et al., 2015*), which disrupts the C-terminal RHR motif, retained a tetrameric structure based on SEC-MALS analysis (*Figure 2A*) and produced very similar results to the wild type protein in the AFM (*Figure 2B and D*). In contrast, the CtIP[L27E] mutant, which disrupts the N-terminal tetramerization interface (*Davies et al., 2015*), ran as an apparently smaller particle in SEC-MALS with predicted molecular weights across the peak ranging from ~ 270 to 190 kDa (most similar to a dimer molecular weight). Importantly, when imaged using AFM, the CtIP[L27E] preparation was completely devoid of the largest class of particle (Class V;

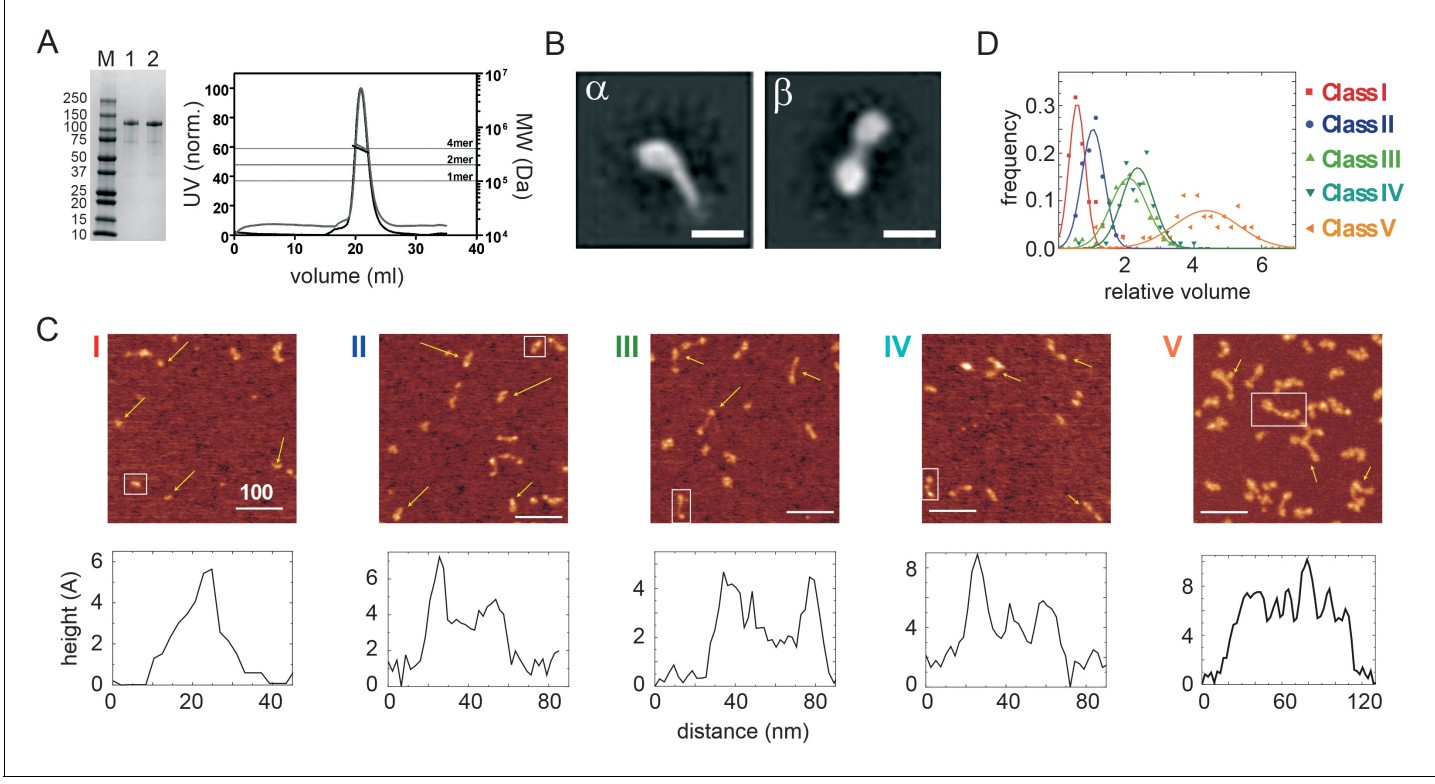

**Figure 1.** Wild type CtIP is a tetrameric protein that forms a dumbbell-shaped particle. (**A**) SDS-PAGE and SEC-MALS analysis of wild type CtIP as prepared (lane 1, black trace) and following dephosphorylation post-purification (lane 2, grey trace). Horizontal lines on the SEC-MALS graph show the expected molecular weights for monomeric, dimeric and tetrameric CtIP species. (**B**) 2D class averages for wild type CtIP observed by negative stain EM. Scale bar = 20 nm. Further examples are shown in the Figure Supplements. (**C**) AFM imaging of purified wild type CtIP protein. Five classes of particles (**I–V**) were detected based on morphology and volume analysis (see text for details). Scale bar = 100 nm. Examples are marked with yellow arrows and further images at higher magnification are shown in the Figure Supplements. Representative height profiles are shown which are derived from the boxed particle. (**D**) Relative volume distributions for the five AFM particle classes shown in C. The volume is determined using a technique in which a fiducial DNA marker is used as a reference (*Fuentes-Perez et al., 2012*). There exists a linear relationship between the relative volume and absolute molecular weight for a wide range of proteins. The molecular weight of the large class V particle imaged here is 380 kDa (i.e. close to the value expected for a tetramer) based on a mean relative volume of 4.4x the marker.

DOI: https://doi.org/10.7554/eLife.42129.002

The following figure supplements are available for figure 1:

**Figure supplement 1.** Negative stain electron microscopy.
DOI: https://doi.org/10.7554/eLife.42129.003

**Figure supplement 2.** Further examples and structural interpretation of AFM particle classes.
DOI: https://doi.org/10.7554/eLife.42129.004

**Figure supplement 3.** The rod between the globular domains is flexible and variable in length.
DOI: https://doi.org/10.7554/eLife.42129.005

**Figure supplement 4.** CtIP purified from insect cells is hyper-phosphorylated.
DOI: https://doi.org/10.7554/eLife.42129.006

*Figure 2C and E*) which, in combination with the SEC-MALS results, suggests again that the class V particle represents tetrameric CtIP. Possible structural interpretations of the other CtIP particle classes we have observed using AFM are shown alongside further examples in *Figure 1—figure supplement 2*.

For all particle classes, the distance between the highest points of the connected dots was on the order of 30 nm, although significant variability was observed consistent with flexibility in the connecting rod and/or the central region of the protein which is predicted to be disordered (see *Figure 1—figure supplement 3* for examples and Discussion).

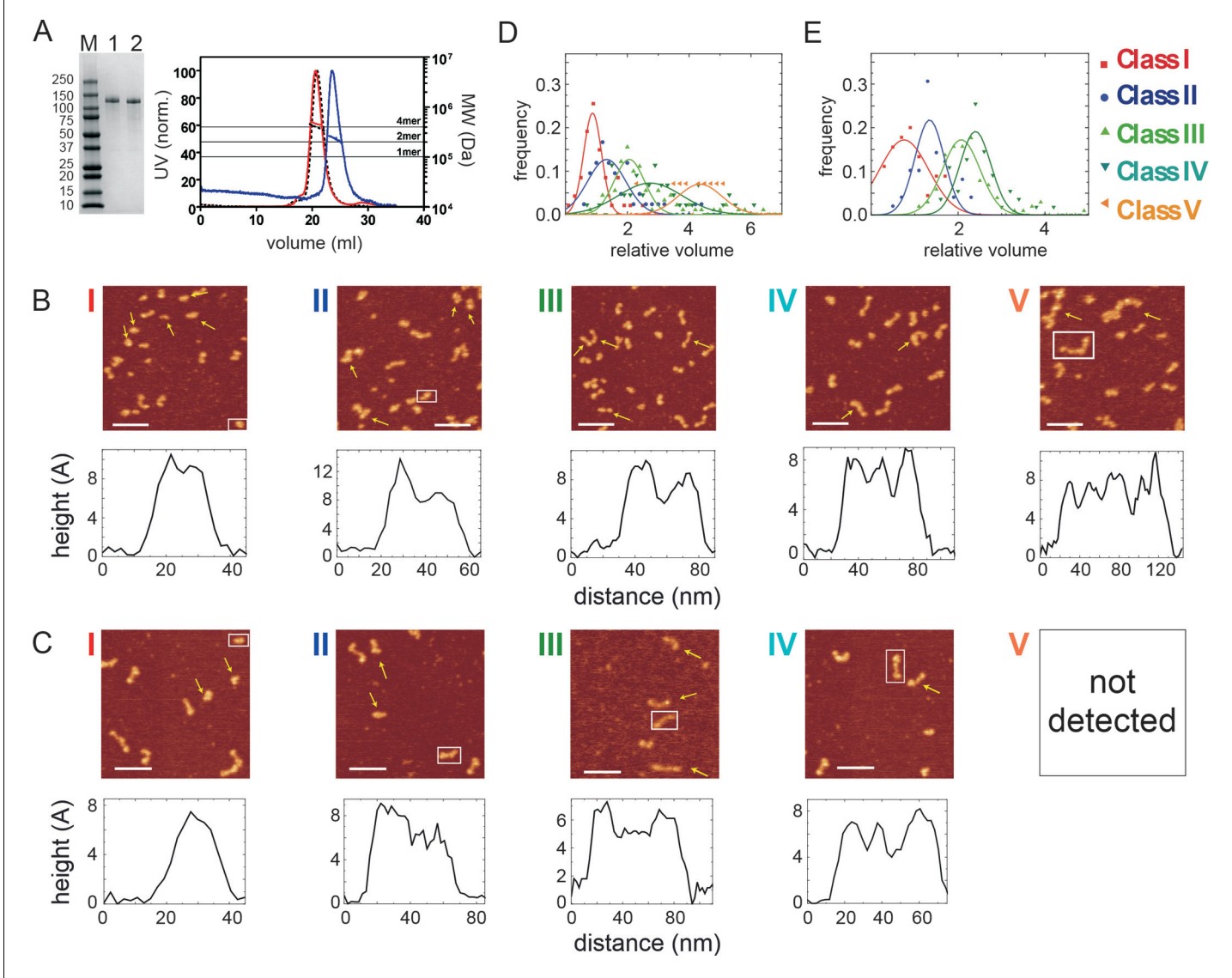

**Figure 2.** Mutation in the N-terminal coiled-coil domain prevents CtIP tetramerisation. (A) SDS-PAGE and SEC-MALS analysis of CtIP L27E (lane 1, blue trace) and CtIP R839A (lane 2, red trace). Data for wild type CtIP is also shown for comparison (dotted black lines). (B) AFM imaging of purified CtIP R839A protein. Five classes of particles (I–V) were detected based on morphology and volume analysis. Representative height profiles are shown below the images which are derived from the boxed particle. (C) AFM imaging of purified CtIP L27E protein. Four classes of particles (I–IV) were detected based on morphology and volume analysis. Representative height profiles are shown which are derived from the boxed particle. (D) Relative volume distributions for the AFM particle classes detected in the CtIP R839A and (E) CtIP L27E preparations. Note that the class V particle is completely absent from the CtIP L27E preparation.

DOI: https://doi.org/10.7554/eLife.42129.007

## The CtIP tetramer binds tightly to DNA in a manner dependent on both the N-terminal tetramerization and C-terminal DNA binding domains

We next used electrophoretic mobility shift assays (EMSA) to compare the binding of CtIP to different DNA substrates (*Figure 3*). CtIP bound tightly to a forked DNA molecule containing a 25 bp duplex region flanked by 20 base ssDNA tails (*Figure 3A*). The binding to a completely single-stranded DNA substrate was weaker and resulted in the formation of more poorly defined complexes in the gel. Binding to a completely duplex DNA substrate was considerably weaker than to a forked substrate. To facilitate a more quantitative approach, we also developed a fluorescence anisotropy assay using HEX-labelled DNA substrates. Firstly, we analysed directly the binding of

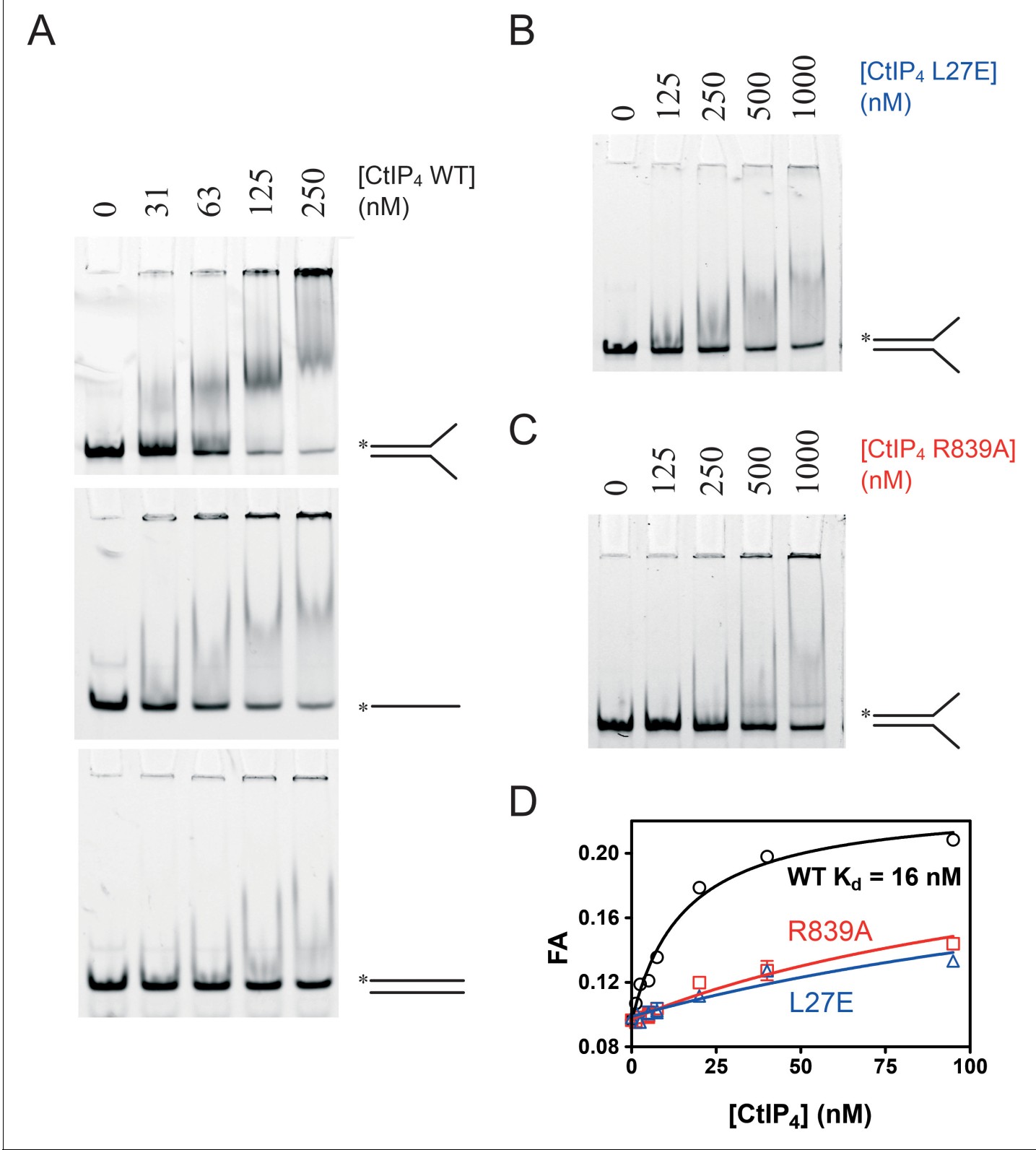

**Figure 3.** CtIP binds preferentially to ss-dsDNA Y-junctions in a manner dependent on both the N-terminal tetramerisation and C-terminal DNA binding motifs. (**A**) Electrophoretic mobility shift assay. Radiolabelled DNA molecules with the different structures (indicated) were incubated with increasing concentrations of CtIP tetramer and run on non-denaturing PAGE gels as described in the Materials and methods section. (**B**) Binding of the

*Figure 3 continued on next page*

*Figure 3 continued*

fork DNA substrate by the L27E mutant protein. (C) Binding of the fork DNA substrate by the R839A mutant protein. (D) Quantitative comparison of wild type and mutant CtIP proteins binding to a HEX-labelled DNA fork structure using fluorescence anisotropy.

DOI: https://doi.org/10.7554/eLife.42129.008

CtIP to a HEX-labelled DNA fork using anisotropy (*Figure 3D*). The binding isotherm was well fit to a hyperbola yielding a value for the equilibrium dissociation constant ($K_d$) of 16 nM, which is somewhat tighter than suggested by using the semi-quantitative EMSA technique with the same substrate. Comparative analysis of the mutant proteins CtIP$^{R839A}$ and CtIP$^{L27E}$ using both assays showed that they were severely defective for DNA binding ($K_d$ = 160 nM and 230 nM respectively) (*Figure 3B–D*).

## Dephosphorylation potentiates the binding of the CtIP tetramer to DNA, allowing determination of the binding stoichiometry

Phosphorylation is crucial for activation of CtIP during the S and G2 phases of the cell cycle. Somewhat surprisingly however, CtIP treated with phosphatases has shown evidence for an *increased* DNA binding affinity although this effect was not extensively investigated (*Anand et al., 2016*). This implies a possible regulation of DNA binding by phosphorylation at one or more undefined sites. Analysis by mass spectrometry showed that our standard CtIP preparation is hyper-phosphorylated (see *Figure 1—figure supplement 4* for details). Therefore, to test how phosphorylation status affects DNA binding quantitatively, we next prepared dephosphorylated CtIP (CtIP$^\lambda$) by treatment with λ phosphatase (*Figure 4*). CtIP$^\lambda$ retained a tetrameric state (*Figure 1A*) and bound to forked DNA significantly more tightly than did untreated CtIP as judged crudely by EMSA (*Figure 4A*). Furthermore, it displayed > 10 fold tighter affinity for forked DNA based on fluorescence anisotropy measurements (*Figure 4B*). The best fit to these data using the tight-binding equation (*Equation 5*) gave $K_d$ = 1.5 nM and stoichiometry (n) = 0.58 CtIP$_4$ per DNA fork. Note that this $K_d$ value should be regarded as an upper limit, because the binding is too tight to measure accurately at the lowest probe concentration we used here (5 nM). The enhanced affinity observed for dephosphorylated CtIP is not the result of an artefactual change in solution conditions imposed by the λ phosphatase or its reaction buffer as the increase in binding affinity can be time-resolved, occurring slowly after the addition of the phosphatase (*Figure 4—figure supplement 1*). The molecular basis for this increased affinity presumably relates to the phosphorylation status of one or more amino acids in CtIP (see Discussion section). Although this very tight binding precluded an accurate determination of $K_d$, it instead allowed us to rigorously investigate the stoichiometry of the CtIP-DNA interaction by performing anisotropy assays with different DNA concentrations, all of which were significantly above the value of the dissociation constant (*Figure 4C*). Under such conditions, the binding isotherms are approximately linear until saturation for a range of DNA concentrations as would be expected. The binding stoichiometry was determined as 0.56 CtIP$_4$ per DNA fork molecule by global fitting of the data to the tight-binding equation (*Equation 5*). This is equivalent to two DNA molecules bound to each CtIP tetramer at saturation (assuming that our CtIP preparation is 100% active). Note that our data is equally as consistent with a CtIP dimer binding to one DNA molecule

## CtIP binds preferentially to DNA substrates with blocked DNA ends

Several lines of evidence have suggested that CtIP function is especially important for the recognition and repair of DNA breaks containing complex DNA end structures. To explore the biochemical basis for this phenomenon we tested whether the affinity of CtIP for different DNA substrates depended on the structure of the DNA ends. To compare the relative affinity of a wider range of DNA substrates, including those with complex DNA end structures such as DNA conjugated to other proteins, we performed competition unbinding assays using fluorescence anisotropy (*Figure 5* and *Figure 5—figure supplement 1–3*; further details for the structures of the DNA molecules used and their construction can be found in the *Supplementary file 1*). CtIP was equilibrated with the HEX-fork DNA at a concentration equivalent to 3 x $K_d$ yielding a high starting anisotropy signal. We then titrated increasing concentrations of unlabelled competitor DNA molecules and monitored the steady-state anisotropy (*Figure 5A*). For each competitor DNA tested, the loss of anisotropy was

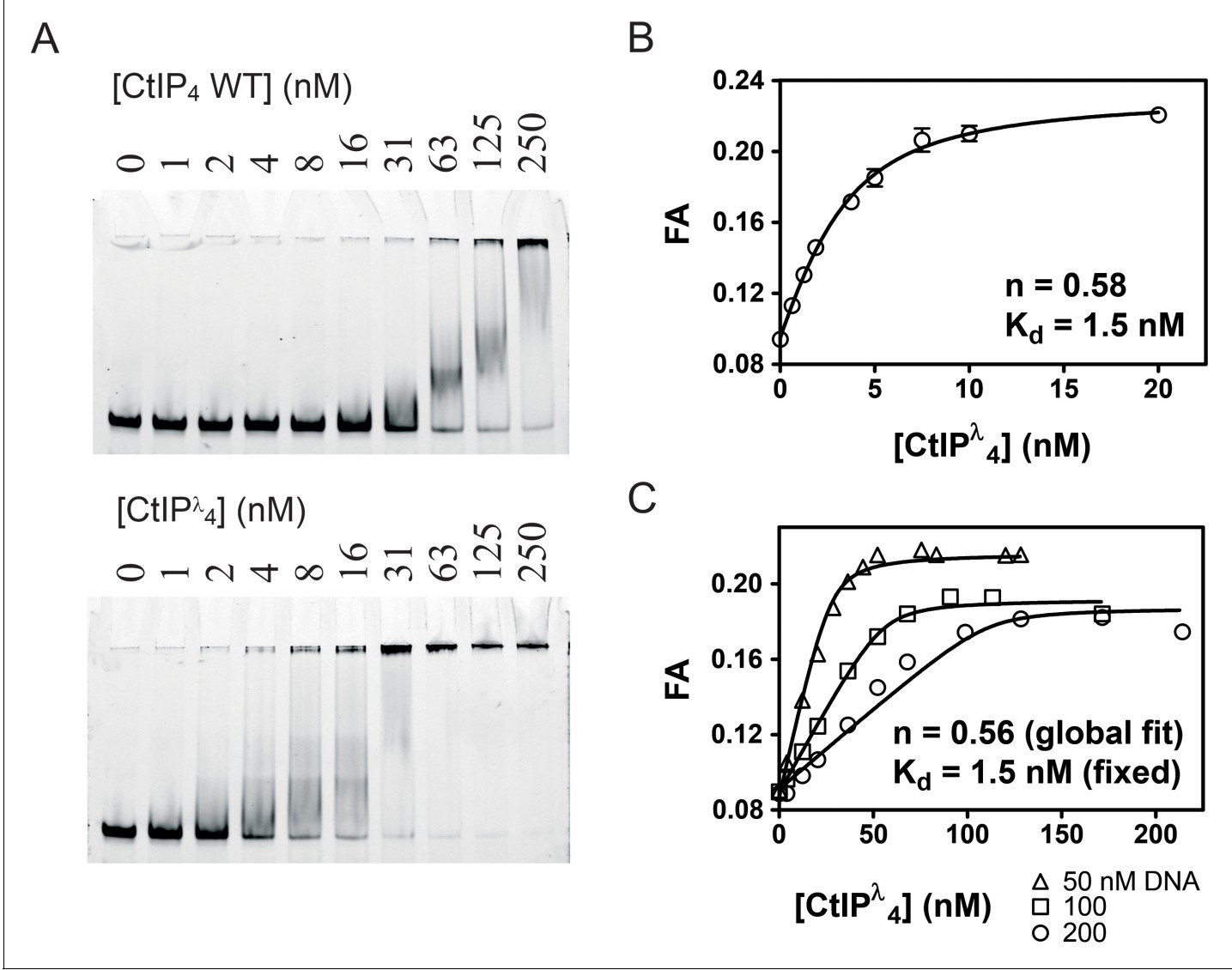

**Figure 4.** Dephosphorylation of CtIP potentiates DNA binding and facilitates determination of the DNA binding stoichiometry. (**A**) EMSA assays comparing the binding of wild type CtIP as prepared and following treatment with λ phosphatase (denoted CtIP^λ) as described in the Materials and methods. (**B**) Fluorescence anisotropy assay monitoring the binding of CtIP to 5 nM HEX-labelled fork DNA. The data were fit to the tight binding equation. (**C**) Assay as in B, but using 50, 100 and 200 nM HEX-labelled fork DNA as indicated. The data were fit globally to the *Equation 5* to determine the binding stoichiometry (n) which measures the number of CtIP tetramers bound to each DNA fork.

DOI: https://doi.org/10.7554/eLife.42129.009

The following figure supplement is available for figure 4:

**Figure supplement 1.** Real time measurement of DNA binding potentiation upon treatment of CtIP with λ phosphatase.

DOI: https://doi.org/10.7554/eLife.42129.010

well fit to a hyperbolic unbinding curve (*Equation 4*) yielding an $IC_{50}$ value; the concentration of competitor at which the binding signal is reduced by 50%. This is consistent with a simple competition between the labelled probe DNA and the competitor DNA for the CtIP DNA binding loci. Note that the measured $IC_{50}$ value is limited at low values by the $K_d$ of the interaction between CtIP and the reference DNA (*Huang, 2003*). Therefore, this method is most useful for comparing the relative affinities of DNA substrates that bind weaker to CtIP than does the reference (HEX-fork) probe, as is generally the case here. Accordingly, the difference in affinity between two competitor DNAs is larger than the difference implied by their $IC_{50}$ values, especially if the values approach the $K_d$ for the probe.

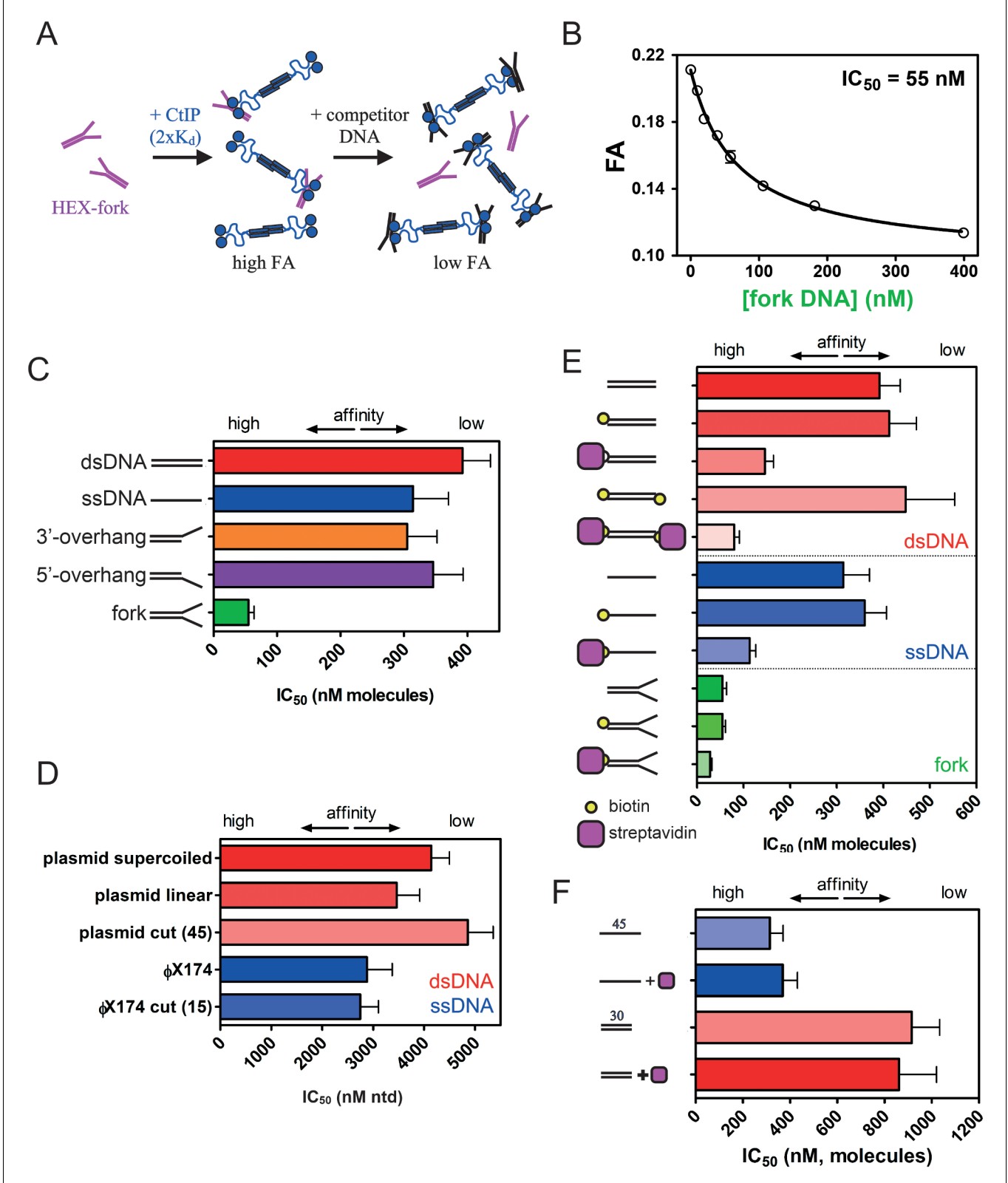

**Figure 5.** CtIP binding to DNA is stabilised by DNA Y-junctions and DNA end blocks. (**A**) Principle of the competition DNA unbinding assay monitored by fluorescence anisotropy. HEX-labelled and unlabelled DNA fork molecules are shown in purple and black respectively. CtIP is in blue. (**B**) Increasing concentrations of unlabelled fork DNA were titrated into a pre-formed complex between CtIP and a HEX-labelled DNA. The data were fit to a hyperbolic decay curve to yield an $IC_{50}$ value for the inhibition of binding. (**C**) Comparison of the $IC_{50}$ values for different DNA structures. Full details of

*Figure 5 continued on next page*

*Figure 5 continued*

the DNA sequences and structures used in these experiments can be found in the Supplementary Information. (D) Comparison of $IC_{50}$ values for single- and double-stranded DNA competitors with different topologies and containing different densities of DNA breaks (bracketed number). Note that the $IC_{50}$ value is quoted in terms of the total nucleotide concentration. (E) Comparison of the $IC_{50}$ values for duplex (red), single-stranded (blue) and forked (green) DNA substrates with and without the ends blocked by biotin:streptavidin. (F) Comparison of $IC_{50}$ values for single- or double-stranded DNA substrates which lack biotin, determined both in the presence and absence of streptavidin. Details for the substrate construction and data for a wider range of substrates are available in the Supplementary Information.

DOI: https://doi.org/10.7554/eLife.42129.011

The following figure supplements are available for figure 5:

**Figure supplement 1.** Structures of competitor DNA molecules used in this study.

DOI: https://doi.org/10.7554/eLife.42129.012

**Figure supplement 2.** Raw data for all $IC_{50}$ measurements.

DOI: https://doi.org/10.7554/eLife.42129.013

**Figure supplement 3.** $IC_{50}$ values for CtIP binding measured for a wider range of competitor DNA molecules.

DOI: https://doi.org/10.7554/eLife.42129.014

**Figure supplement 4.** CtIP and Ku display distinctive DNA binding modes.

DOI: https://doi.org/10.7554/eLife.42129.015

In our first competition experiment, we found that titration of the unlabelled fork DNA against the reference probe yielded an $IC_{50}$ value of 55 nM (*Figure 5B*). In comparison, fully single- and double-stranded DNA substrates were relatively poor competitors ($IC_{50}$ values of 314 nM and 392 nM respectively) in qualitative agreement with the EMSA analysis which had also shown a clear preference for forked DNA (*Figure 5C*). Interestingly, this effect required the presence of both the 3'- and 5'- ssDNA overhangs in the fork (i.e. a Y-junction) as neither polarity overhang alone significantly improved the observed binding relative to duplex (*Figure 5C*). We next titrated a variety of much larger competitor DNA molecules which differed in terms of their average length, topology, and whether they were single- or double-stranded (*Figure 5D*). In terms of the total nucleotide concentration, the $IC_{50}$ value was broadly similar for single- and double-stranded DNA. Moreover, the $IC_{50}$ was largely unaffected by whether the DNA was circular, linearised or fragmented into many pieces. This demonstrates that CtIP does not bind preferentially to DNA ends compared to internal sites because the efficacy of the competitor is independent of the total concentration of free DNA ends. Further experiments with a range of short single-stranded oligonucleotides showed that binding was length dependent (in terms of absolute molecule concentrations) and very poor on substrates shorter than around 30 nucleotides (*Figure 5—figure supplement 3*).

Next, to investigate how modified 'complex' DNA ends might affect the affinity of CtIP for DNA, we end-labelled different DNA competitor molecules with a model nucleoprotein block based on the biotin:streptavidin interaction (*Figure 5E*). The addition of biotin labels alone to the termini of the DNA competitor molecules had no significant effect or slightly weakened the $IC_{50}$ values. However, the additional binding of streptavidin to the terminal biotin moieties dramatically increased the efficacy of the DNA molecules as competitors. For example, a fully duplex 45 base pair DNA molecule displayed an $IC_{50}$ value of 392 nM and biotinylation of either one, or both of the ends, resulted in values of 413 and 448 nM respectively. However, the $IC_{50}$ value reduced to 146 nM with a single biotin:streptavidin block and further to 80 nM with a biotin:streptavidin block at either end (compare the red bars in *Figure 5E*). Moreover, similar effects were observed on both single-stranded and forked DNA substrates (blue and green bars respectively; *Figure 5E*). The enhanced binding afforded by streptavidin blocking was apparent regardless of whether the biotin moiety was placed at the 5' or 3' end of DNA strands (*Figure 5—figure supplement 3*). In these competition experiments, streptavidin was added at a large excess (8.5-fold) to prevent multimerization of the competitor DNA. Importantly, the addition of streptavidin to either single- or double-stranded competitor DNAs that lacked biotin moieties had no effect on the observed $IC_{50}$ value (*Figure 5F*). This shows that the observed effect is dependent on binding of a streptavidin molecule to the biotinylated DNA end, as opposed to a non-specific effect of free streptavidin. Nevertheless, it is difficult to exclude the possibility that the effect observed here is due to a weak non-specific interaction between CtIP and streptavidin. In an attempt to address this, we also found that when a biotin was placed in the middle of a duplex DNA molecule, the $IC_{50}$ value was now *weaker* in the presence of streptavidin

(*Figure 5—figure supplement 3*). This could indicate either that the streptavidin is blocking the association of CtIP with internal sites and/or that the positive effect of streptavidin binding on CtIP affinity requires it to be positioned specifically at a DNA end.

The *enhancement* of DNA binding we observe as a result of DNA end blocking by streptavidin also serves to highlight the distinctive and unusual DNA binding properties of CtIP in comparison to more conventional DSB repair factors. For example, the association between DNA and the NHEJ-factor Ku (with which CtIP competes to determine pathway choice in DSB repair) is seen to be severely inhibited by biotin:streptavidin end blocks in experiments analogous to those described above (*Figure 5—figure supplement 4*).

## CtIP bridges DNA in vitro

We next used atomic force microscopy to directly image CtIP:DNA interactions and potential bridging interactions. We engineered a DNA substrate containing short ssDNA tails at both ends because EMSA assays had shown that this was a favourable structure for binding (see Materials and methods for details). Initially we used AFM to image the DNA substrate alone (*Figure 6A*). The DNA was homogenous, consisting virtually entirely of molecules with a single contour length as would be expected (i.e. the DNA substrate was monomeric). The length of the substrate DNA with single-strand overhangs is 587 bp, which we expect to be ~ 200 nm long based on a value of 0.34 nm/bp for B-DNA. In good agreement with this estimate, the mean contour length measured for the control bare DNA substrate was 195 ± 22 nm. We next repeated this imaging experiment, but in the presence of increasing concentrations of CtIP. In the presence of CtIP we observed free CtIP as well as CtIP:DNA complexes, albeit at a lower frequency than might be expected given the very tight binding of CtIP to DNA in free solution (*Figure 6B* and *Figure 6—figure supplement 1*). This might reflect the need to treat the mica surface with $Mg^{2+}$ ions to facilitate DNA deposition, a condition we know to be inhibitory to CtIP:DNA interaction (see Materials and methods for details). Furthermore, we saw an increase in the number of DNA molecules with contour lengths equal to 2x, 3x or even 4x the monomer length (*Figure 6B–D*). This is illustrated in *Figure 6C*, which shows the cumulative frequency of DNA molecules as a function of DNA length for increasing doses of CtIP. Considering a threshold length of 195 + 22 nm (mean control length + SD), the percentage of molecules above this threshold is 4% in the 'no-protein' control experiment rising to about 30% in samples containing the highest concentrations of CtIP. In terms of the bridged molecules which are exactly the expected lengths (i.e. multiples of 195 nm within 2σ error) we see none whatsoever for the 'no protein' control, rising to ~ 7% in samples containing wild type CtIP (*Figure 6D*). Note that this quantification method does not score for intramolecular bridging events as these would result in single length circularised products, although these were observed in our experiments (*Figure 6—figure supplement 2*). As would be expected based on our observation that the binding specificity is not strictly for ends, we saw a range of intermolecular connections between two DNA duplexes including end-to-end, end-to-centre and centre-to-centre (*Figure 6B*, zoomed-in panels). This observation serves to highlight the fact that DNA bridging cannot be the result of sticky end annealing between DNA substrates lacking the ligated forked ends. Moreover, in many cases, analysis of the height and width of the bridging interfaces between DNA molecules provided evidence for additional mass which we interpret as CtIP (see *Figure 6—figure supplement 1* for data). We did observe instances of DNA bridges with no additional mass. These might reflect CtIP dissociation from DNA concomitant with deposition onto the $Mg^{2+}$-treated mica surface as discussed above. In the presence of CtIP, a small proportion of the DNA substrate was less than a single contour length, and many of the bridged molecules were not (within error) multiples of single contour lengths as might be expected. This could be explained by the presence of a weak nuclease activity in the preparation (see *Figure 6—figure supplement 3* for further discussion and analysis of this activity), and/or by CtIP binding causing a condensation effect on the DNA such that contour length is underestimated.

We next compared the bridging ability of dephosphorylated CtIP and the two mutant proteins L27E and R839A, as well as performing an additional negative control with BSA in place of CtIP (*Figure 6D*; *Figure 6—figure supplement 4*). In comparison to wild type, dephosphorylated CtIP was more effective at bridging DNA molecules. Both the total percentage of bridged molecules and the proportion of the bridged molecules displaying greater than 2x contour length were increased. In contrast, both mutant proteins were severely defective in DNA bridging as would be expected based on their weak DNA binding affinities. On this measure, no bridging was observed in the 'no-

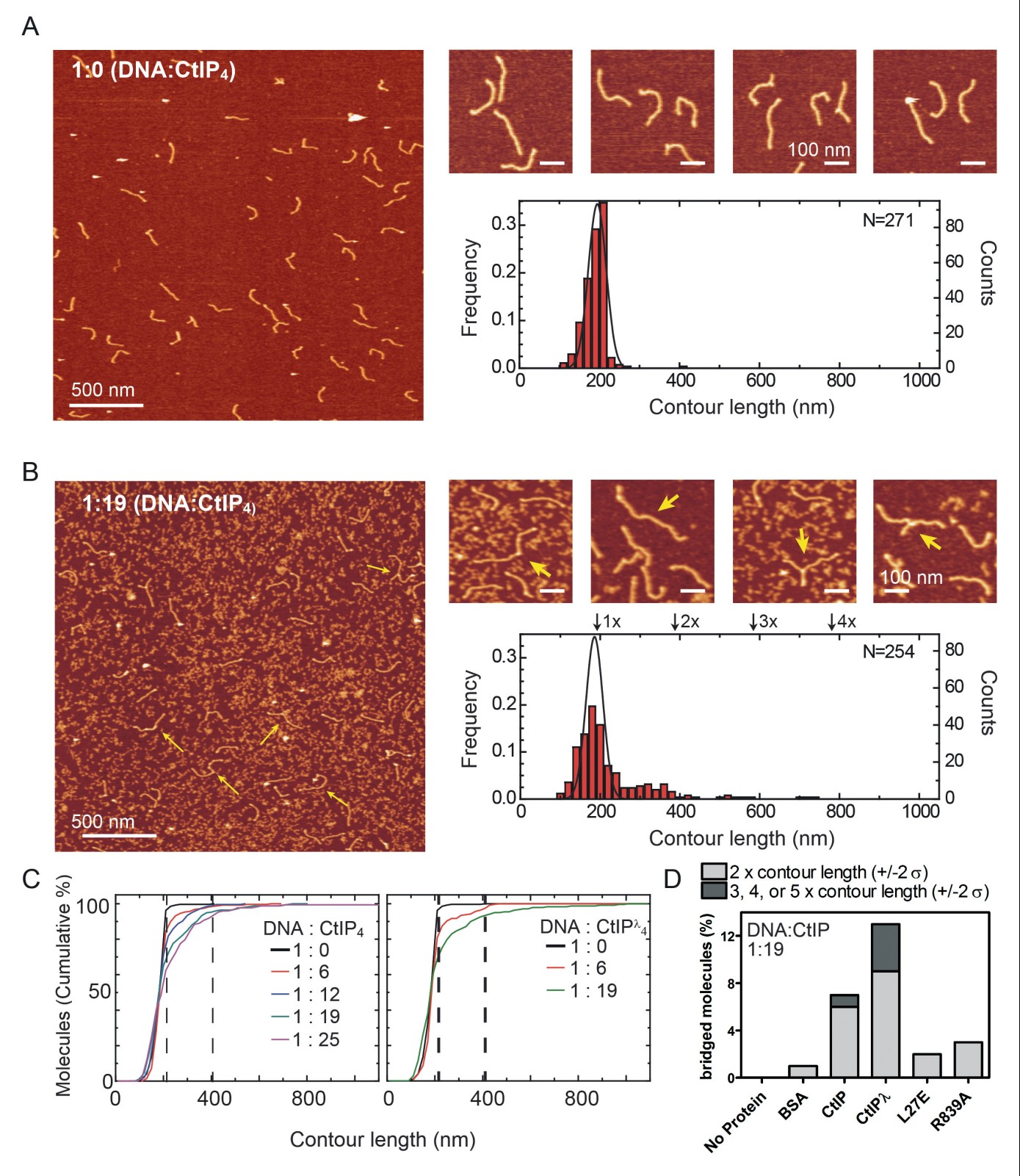

**Figure 6.** CtIP promotes intermolecular DNA bridging. (**A**) Representative AFM images of forked DNA substrates in the absence of CtIP. The contour length histogram shows a single gaussian peak centred on a value equivalent to a single contour length. (**B**) Representative AFM images of forked DNA substrates in the presence of CtIP. The contour length histogram shows multiples peaks for single-, double-, triple and even quadruple contour length peaks (indicated). The solid line shows the gaussian fit for data collected in the absence of CtIP for comparison. Note that the size of the single-contour

*Figure 6 continued on next page*

*Figure 6 continued*

peak is substantially reduced. The zoomed-in panels show examples of bridged DNA molecules (yellow arrows). (C) Cumulative frequency of DNA molecules longer than a given contour length, highlighting the CtIP-dependent increase in 2mer and 3mer DNA substrates. The vertical dashed lines are equivalent to theoretical single- and double- contour lengths. (D) Histogram showing percentage of bridged molecules for wild type, dephosphorylated and mutant CtIP proteins at a single fixed CtIP concentration. The light grey data shows the percentage of molecules (within error) at 2x contour length, whereas the dark grey data is the sum of the percentage of molecules at 3x, 4x and 5x contour lengths.

DOI: https://doi.org/10.7554/eLife.42129.016

The following figure supplements are available for figure 6:

**Figure supplement 1.** Evidence for CtIP bound to DNA at bridging interfaces.
DOI: https://doi.org/10.7554/eLife.42129.017

**Figure supplement 2.** CtIP promotes intramolecular DNA bridging.
DOI: https://doi.org/10.7554/eLife.42129.018

**Figure supplement 3.** Assessment of nuclease activity in CtIP preparations.
DOI: https://doi.org/10.7554/eLife.42129.019

**Figure supplement 4.** Dephosphorylated CtIP promotes intermolecular DNA bridging.
DOI: https://doi.org/10.7554/eLife.42129.020

protein' control, although a second negative control using BSA did show a low level of apparent activity. This suggests that the activity observed for the mutant proteins partially reflects a crowding effect in addition to any *bona fide* DNA bridging.

Interestingly, bridging experiments performed with a substrate that lacked the terminal ssDNA forks showed a reduced level of bridging, suggesting that the tighter DNA binding afforded by modifications to the DNA ends also facilitates DNA bridging as would be expected (data not shown). We also attempted to measure DNA bridging by exploiting a pulldown assay that has been used to study intermolecular DNA bridging by the *S. pombe* orthologue Ctp1 (*Andres et al., 2015*). However, we found that human CtIP bound non-specifically to the beads used for the pulldown under a wide range of conditions, and so we were unable to distinguish between DNA binding and bridging activity using that assay (data not shown).

## Discussion

In this study we purified human CtIP protein and characterised its structure and DNA binding properties in vitro. The purified protein is tetrameric and adopts a striking 'dumbbell' architecture, which we have observed with both negative-stain EM and AFM. This shape is expected based on the hypothesis that CtIP adopts a 'dimer of dimers' arrangement, in which the tips of long N-terminal parallel coiled coils interact to form the tetrameric interface observed in crystal structures (*Andres et al., 2015*; *Davies et al., 2015*; *Forment et al., 2015*) (*Figure 7*). We found that CtIP bound tightly to DNA using both EMSA and fluorescence anisotropy assays. Evidence that a major DNA binding locus resides in the distal C-terminal domains is provided by the inability of the R839A mutant to bind effectively to DNA, despite retaining a normal oligomeric state (this work; *Andres et al., 2015*; *Figure 7*). The globular structure of the C-terminal domains and the 2:4 stoichiometry of DNA to CtIP monomers implies a dimeric arrangement of the DNA binding domains. It will be of particular interest to determine a high-resolution structure for this region of the protein bound to DNA as, despite the presence of the small 'RHR' motif, this region of the protein does not obviously resemble any known DNA binding domain.

A surprising result was that the L27E point mutant also displayed a dramatically reduced DNA binding affinity. This mutation was designed on the basis of crystal structures of the extreme N-terminal coiled-coil to disrupt protein:protein interactions responsible for tetramer formation, and renders CtIP non-functional in vivo (*Davies et al., 2015*). Indeed, analysis by both SEC-MALS and AFM suggested that this mutant was completely unable to form tetrameric assemblies. Our work suggests therefore, that mutations in the tetramer interface may have a more broadly destabilising effect on the biochemical activity of CtIP than simply interfering with the oligomeric state. It may be that association with DNA requires communication between two distant binding loci, although we have observed no evidence for co-operativity in our binding experiments. Alternatively, the failure to

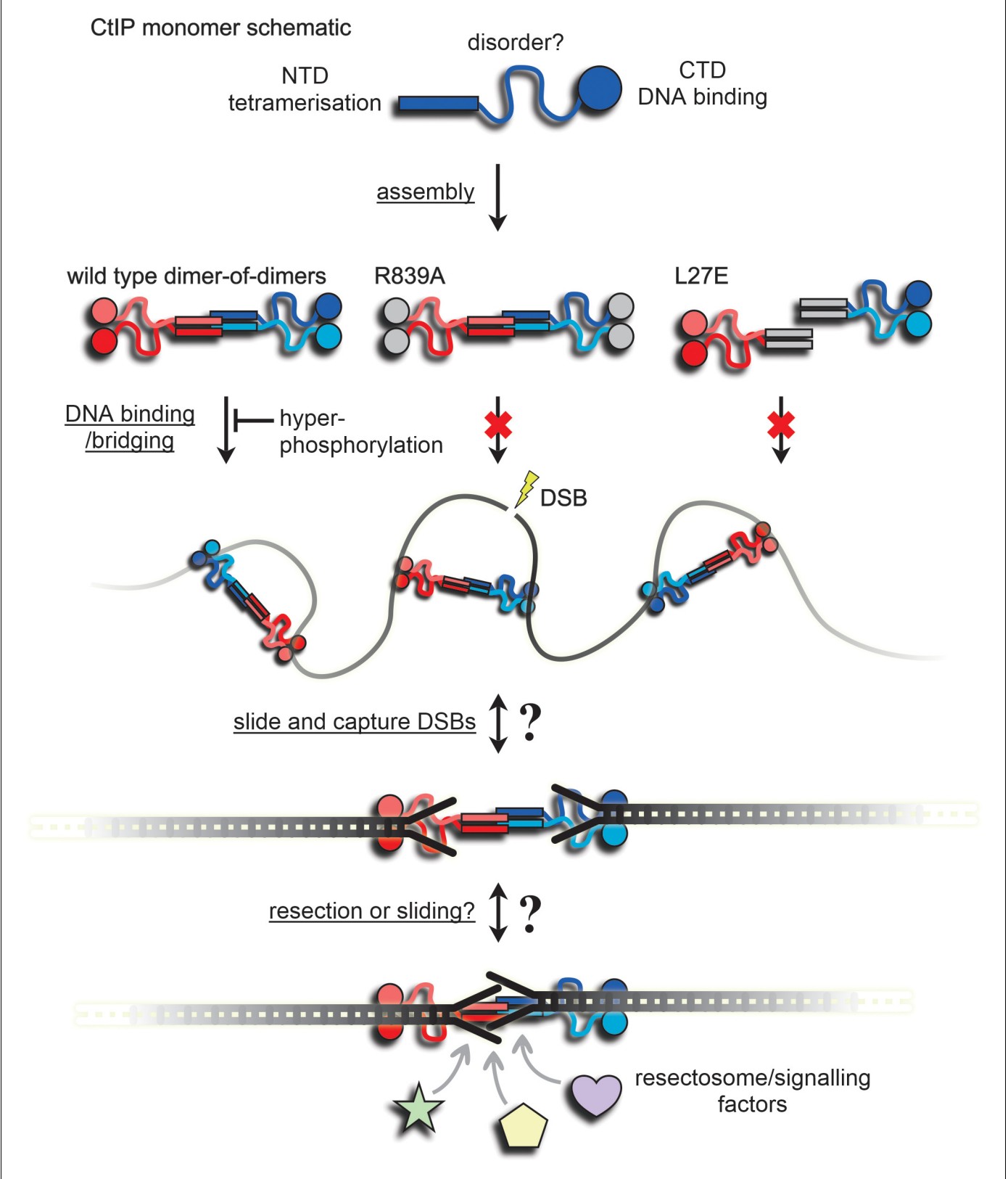

**Figure 7.** A speculative model for binding and bridging of broken DNA ends by CtIP. The CtIP monomer comprises at least three functional regions; an N-terminal tetramerization domain, a central region of predicted disorder, and a C-terminal DNA binding domain. This assembles to form a dumbbell-shaped tetramer (evidence in *Figure 1*). Wild type CtIP binds tightly to two DNA molecules via C-terminal RHR motifs, but this is disfavoured

*Figure 7 continued on next page*

*Figure 7 continued*

by hyperphosphorylation (*Figures 3* and *4*). Mutation within this motif (R839A) prevents DNA binding and bridging (*Figures 3* and *6*). Mutation within the N-terminal coiled-coil domains prevents assembly into a tetramer, DNA binding and DNA bridging (*Figures 2*, *3* and *6*). The grey coloured domains are intended to indicate their inactivation by mutation. The inability of L27E to bind to DNA implies that mutations in the coiled coils have effects on the distal DNA binding domains (see main text for discussion). Wild type CtIP binds DNA without preference for ends per se (i.e. internally) and bridges DNA segments either intra- and inter-molecularly (*Figures 5* and *6*, and the Figure Supplements). Based on the preferential binding to DNA with forked or blocked ends (*Figure 5*), we *speculate* that CtIP may then slide and capture DNA ends possessing forked structures or nucleoprotein conjugates, because they prevent dissociation from the free ends (see main text for discussion). This arrangement might then allow the dimeric sub-assemblies within CtIP to slide away from each DNA end to which they are bound, facilitating resection and/or access to the DSBs for resection and signalling factors while maintaining their pairing. This proposal will be the subject of future work.

DOI: https://doi.org/10.7554/eLife.42129.021

tetramerise may simply have an unfavourable effect on the global architecture of CtIP that disfavours DNA binding.

There was no evidence to suggest that CtIP bound preferentially to DNA ends in vitro, as might be expected given its well-defined role in the early stages of DSB repair. However, we did find that modifications to the ends of DNA substrates, including ssDNA forks and model nucleoprotein blocks, strongly increased their affinity for CtIP. This provides a biochemical basis for the observation that the activity of CtIP is especially important for the repair of 'complex' or 'dirty' DSBs in vivo, such as those programmed during meiosis by Spo11 or formed during topoisomerase poisoning by etoposide (*Aparicio and Gautier, 2016*). It can also help to explain why MRN-dependent nicking of DNA neighbouring blocked DNA ends requires (or is at least strongly stimulated by) CtIP, which interacts physically and functionally with MRN (*Anand et al., 2016*; *Williams et al., 2009*).

The mechanism by which modifications to DNA ends increase the affinity of CtIP for DNA remains to be determined. One simple possibility would be that CtIP interacts directly with the protein or nucleic acid conjugates that we have appended to our DNA substrates. However, we do not favour such a model: it is difficult to see how a direct protein-mediated interaction could recognise the fundamentally different types of ends that we have used in vitro or which must presumably be recognised in vivo. Alternatively, the protein might act via a 'slide and capture' mechanism in which the interaction is initially with internal sites with the DNA ends acting as exit sites. In that case the dissociation may be disfavoured if the end is modified with any bulky blocking moiety, including Y-junctions and nucleoprotein complexes, such that complex ends would act to capture CtIP (*Figure 7*). Presumably, the localisation of CtIP at DSBs in vivo is also mediated by the many other proteins with which it interacts, including the BRCA1-BARD1 complex and MRN, or by the post-translational modifications which they facilitate.

CtIP was found to be hyperphosphorylated as prepared from insect cells lysed in the presence of dephosphorylation inhibitors. This is as expected based on several previous reports of regulatory phosphorylation events on CtIP mediated by kinases linked to the DNA damage response or cell cycle (*Huertas and Jackson, 2009*; *Peterson et al., 2013*; *Wang et al., 2013*). Intriguingly, we observed a strong potentiation of DNA binding affinity (>10 fold tighter) when CtIP was dephosphorylated either during- or post-purification. This is consistent with a previous report that highlighted an apparent stimulation of DNA binding by dephosphorylation (*Anand et al., 2016*), but unintuitive given that phosphorylation events on CtIP activate it in vivo. It remains unclear whether this effect is manifest by one or a small number of dephosphorylation events at specific residues, and therefore may represent a regulatory mechanism, or whether this simply represents a general effect of removing extensive negative charge from the surface of the protein, and this will be the subject of further study. Nevertheless, the regulation of DSB repair factors by downregulation of DNA binding via post-translational modification is certainly not unprecedented (*Blanco et al., 2014*; *Honda et al., 2011*; *Lee et al., 2016*).

The very tight binding mode of dephosphorylated CtIP facilitated experiments to determine the binding stoichiometry between the CtIP tetramer and short DNA fork substrates. We found that each tetramer can bind two such DNA molecules. Given the dumbbell architecture that we have observed and the location of a key DNA binding motif in the distal C-terminus, we speculate that these two DNA molecules bind to the opposing ends of the tetramer, where a major DNA binding locus might be formed by dimerization of the C-terminal domains including the RHR motif. This

arrangement may help to co-ordinate the repair and eventual re-joining of two DSBs (*Figure 7*). In this respect, it is interesting to note that our AFM experiments observed significant flexibility in the connecting rod between the globular domains, which may help facilitate such co-ordination. Consistent with this model, we also showed that CtIP was able to bridge DNA segments intermolecularly. Such a bridging activity might also help to tether the homology donor DNA in proximity to the broken DNA molecules to promote strand exchange. Given the very large number of reported CtIP interaction partners, it is possible that CtIP acts as a binding platform for co-ordinating two ended DSBs with the recruitment of signalling factors of the DNA damage response and for the assembly of a 'resectosome' complex to promote repair by recombination. Higher resolution structural information for full length CtIP protein and its interactions with partner proteins will be instrumental in understanding its important role in DNA break recognition and the regulation of homologous recombination.

## Materials and methods

### Protein expression and purification

A synthetic gene codon-optimised for *S. frugiperda* (Geneart, Invitrogen) encoding wild type human CtIP was cloned into the pACEBac1 vector using the BamHI and XbaI restriction sites for use in the MultiBac system (Geneva Biotech). This was then screened for expression and purification using affinity tags in different positions. Following cleavage of a construct with a C-terminal 3C-cleavable StrepII tag, we obtained CtIP in good yield and purity. The full length recombinant protein as prepared (i.e. post-3C cleavage) contains N- and C-terminal extensions: MELL- and -SGLEVLFQ respectively (MW = 103171 Da monomer), with the rest of the protein identical to UniProt entry Q99708. A contaminant band of approximately 75 kDa is also routinely present following purification, and this was identified as a CtIP degradation product by mass spectrometry (data not shown). Mutagenesis (Quikchange XL, Agilent) was performed using this construct to create the L27E and R839A mutants. Bacmids were prepared by transposition of these plasmids and were used to transfect Sf9 insect cells in Insect Express media (Lonza) before viral amplification in the same cell line using standard techniques. For large scale expression, 500 mL of Hi5 cells at density $2 \times 10^{6}$/ mL were infected with 25 mL of P3 virus and harvested by centrifugation after 70 hr at 27°C with shaking. The pellets were lysed into buffer containing 50 mM Tris pH8.0, 500 mM NaCl, 2 mM DTT, 10% glycerol, 1 mM $Na_3VO_4$, 20 mM β-glycerophosphate, protease inhibitor cocktail (Roche) and then sonicated on ice for a total of 3 min. After centrifugation at 4°C for 30 mins at 50000 g, the cleared lysate was applied to Streptactin beads (GE Healthcare) in batch and incubated for 1 hr at 4°C with rotation. After washing five times in batch with buffer containing 20 mM Tris HCl pH8.0, 500 mM NaCl, 5% glycerol, 1 mM DTT, the protein was then eluted in the same buffer containing 2.5 mM desthiobiotin. The CtIP-containing fractions were diluted to approximately 125 mM NaCl and applied to a 5 mL Heparin column (GE Healthcare). After washing, CtIP was eluted with a gradient from 150 mM to 1 M NaCl over 20 CV in buffer containing 20 mM Tris HCl pH8.0, 5% glycerol, 1 mM DTT. The CtIP-containing fractions were pooled and digested overnight at 4°C with 40 units of 3C protease to remove the StrepII tag. The cleavage reaction was run over a 5 mL Streptactin column (Qiagen) to remove any uncleaved CtIP and free StrepII peptide and the cleaved CtIP-containing flow-through collected. This was diluted to give a NaCl concentration of approximately 125 mM and then loaded onto a 1 mL MonoQ column (GE Healthcare). CtIP was eluted with a gradient from 125 mM to 500 mM NaCl over 20 CV in buffer containing 20 mM Tris HCl pH8.0, 5% glycerol, 1 mM DTT. For some preparations, the most concentrated 1 mL from the MonoQ elution was applied to a Superdex 200 column in buffer containing 20 mM Tris HCl pH8.0, 200 mM NaCl, 5% glycerol, 1 mM DTT, followed by concentration of the CtIP peak using a centrifugal filter unit (Millipore) and storage at −80°C. Protein concentration was determined using a theoretical extinction coefficient of 37360 $M^{-1}$ (monomer) $cm^{-1}$. All CtIP concentrations are stated as tetramer ($CtIP_4$). Analysis of the purified protein by Orbitrap LC-MS/MS spectrometry was performed by the University of Bristol Mass Spectrometry Facility. Purified recombinant human Ku70/80 heterodimer was a gift from Charles Grummit (University of Bristol).

## Preparation of DNA substrates for binding assays

HPLC-purified single-stranded oligonucleotides (Eurofins, ATD Bio) were used as supplied. For dsDNA substrates, oligonucleotides were annealed by heating at 50 µM in 50 mM Tris pH7.5, 150 mM NaCl, 1 mM EDTA for 10 mins at 95°C and allowed to cool slowly overnight. For streptavidin-blocked DNA substrates, a 6 µM biotinylated DNA solution was blocked with 50 µM streptavidin (8.5-fold molar excess) to promote a 1:1 stoichiometry of biotin:streptavidin. For substrates with two biotinylated ends, a 6 µM DNA solution was blocked with 100 µM streptavidin (8.5-fold molar excess). The blocking reactions were incubated for 5 mins at 25°C before performing titrations. The experiments examining the effect of ends on CtIP binding used: φX174 Virion (circular ssDNA), φX174 Virion cut with HaeIII (15 fragments of ssDNA), pSP73JY10 (supercoiled dsDNA, *Yeeles et al., 2011*), pSP73JY10 cut with BamHI (linear dsDNA) and pSP73JY10 cut with Hin1II (45 fragments of dsDNA).

## Size exclusion chromatography coupled to multiple angle light scattering (SEC-MALS)

SEC-MALS was used to determine the absolute molecular masses of full-length CtIP WT and mutants. Approximately 50 µg samples of CtIP were loaded at 0.5 ml/min onto a Superose 6 10/300 size-exclusion chromatography column (GE Healthcare) in 20 mM Tris, pH 8.0, 200 mM NaCl, 1 mM TCEP using an Agilent HPLC. The eluate from the column was coupled to a DAWN HELEOS II MALS detector (Wyatt Technology) and an Optilab T-rEX differential refractometer (Wyatt Technology). ASTRA six software (Wyatt Technology) was used to collect and analyse light scattering and differential refractive index data according to the manufacturer's instructions. Molecular masses and estimated errors were calculated across individual eluted peaks.

## Atomic Force Microscopy

*Sample preparation:* The forks for the ends of the DNA substrate were assembled by annealing two complementary oligonucleotides (Eurofins) at 50 µM in 50 mM Tris pH7.5, 150 mM NaCl, 1 mM EDTA. The forks were designed to have two $dT_{20}$ tails to comprise the forked region (to minimise DNA:DNA interactions in trans), a 25 bp duplex region and a 4-nucleotide overhang compatible with specific restriction endonuclease-generated sticky ends. The 'front' fork had a 3'-overhang (SphI) and the 'end' fork a 5'-overhang (HindIII). Both forks also had a 5'-phosphate group at one end for downstream ligation. 100 µg of plasmid pSP73 JY10 (*Yeeles et al., 2011*) was cut with SphI and HindII (NEB) and the 500 bp band was gel purified (1xTAE 1% agarose). The two forks (10 µM each) and 500 bp fragment (65 nM) were then ligated overnight at 16°C with T4 DNA ligase. The product ran at a position equivalent to a duplex of ~ 600 bp, and was gel purified (1xTAE 1% agarose) away from the starting unligated DNA material with approximately 1 µg being recovered.

*Imaging:* For imaging of protein complexes, the stock protein solution was first diluted in storage buffer (20 mM Tris-HCl pH 8.0, 200 mM NaCl, 1 mM DTT) to a final concentration of 4 nM. Then, the protein solution was deposited onto a freshly cleaved mica surface. After 30 s, the mica surface was washed with 3 ml Milli-Q water (Millipore, Billerica, MA) and gently dried under nitrogen air flow. DNA bridging reactions were performed with 10 nM DNA and increasing concentrations of $CtIP_4$(0–250 nM) for different DNA:$CtIP_4$ ratios (1:0, 1:6, 1:13, 1:19, 1:25) in buffer A (100 mM HEPES 7.5, 100 mM NaCl). The binding reaction mixture was incubated for 15 min in a 2 µl reaction volume. Then, the mica was pretreated with a solution of 7.5 mM $MgCl_2$ and washed with 3 ml MilliQ water. Note that this pre-treatment is essential for successful deposition of DNA and DNA:protein complexes onto the mica surface and also that $MgCl_2$ at moderate concentrations is inhibitory to CtIP binding (data not shown). Residual $Mg^{2+}$ may cause unbinding of CtIP from DNA concomitant with deposition onto the mica, explaining why relatively few DNA:CtIP complexes were observed despite the very tight binding observed in free solution. A droplet of 18 µl of buffer B (4 mM HEPES 7.5, 10 mM NaCl) was added onto the pretreated mica and the 2 µl of binding reaction added to the droplet. Therefore the final solution deposited on the mica was 10 times less concentrated than in the binding reaction (*Fuentes-Perez et al., 2012*). The mica samples were incubated for 30 s, washed with 3 ml MilliQ water, and dried under nitrogen air flow. Samples were imaged in air at room temperature with Point-Probe-Plus tips (PPP-NCH, 42 N/m, 330 kHz, Nanosensors, Neuchâtel,

Switzerland) in amplitude modulation mode AFM (Nanotec Electrónica, Madrid, Spain). Image flattening and in-plane substraction were performed with WSxM freeware (*Horcas et al., 2007*).

*DNA length analysis:* AFM images of DNA molecules in the presence or absence of CtIP protein have been analyzed using the WSxM freeware and Origin Pro 8. A profile along the DNA molecules present in 9–10 images from different samples (typically 250 molecules) was used to obtain the value of the contour length for the analysis. The data from DNA molecules are classified depending on the concentration ratio of DNA with respect to CtIP tetramer and analyzed by histogram representation including data from all the molecules. The histograms were fitted to a Gaussian distribution and the mean contour length and error were extracted from the Gaussian parameters.

*Protein volume characterization with the atomic force microscope:* Direct calculation of volumes by AFM images is affected by a tip convolution effect that may distort the shape and size of the molecules imaged. Correction to the volume measured by the AFM was performed by considering the volume of a fiducial marker, namely a piece of DNA, and following the procedure described in (*Fuentes-Perez et al., 2012*). Images of $1000 \times 1000$ nm$^2$ at $512 \times 512$ pixels were selected from CtIP-DNA samples. The volume of CtIP was calculated from the heights and area of variable size windows depending on the protein size and subsequent subtraction of a basal volume of a similar blank window near the CtIP window. The volume of the fiducial DNA was calculated from a fixed $20 \times 20$ nm$^2$ window. The normalized CtIP/DNA values ('relative volume') were calculated for each protein species and displayed as histograms. Proteins were classified by the overall shape of the molecules in five different classes.

## Negative stain electron microscopy and image analysis

Purified CtIP was diluted to approximately 0.15 mg/ml in 20 mM Tris HCl pH8.0, 1 mM DTT, adsorbed onto glow-discharged carbon-coated grids, and negatively stained with 1% uranyl acetate. The grids were observed using a FEI T12 Spirit electron microscope operating at 120 kV and the images were recorded using FEI 2K eagle camera at a magnification of 52,000. Image processing was performed using RELION 2.1 (*Scheres, 2012*). Initially 15,236 particles were picked from 96 micrographs and reference-free 2D classification was used to remove poor particles, resulting in 6924 particles with approximately a 35/65 ratio divided between the two classes α and β.

## Electrophoretic mobility shift assays

5′-Cy5-labelled DNA substrates (2.5 nM final) were mixed with increasing amounts of CtIP protein (up to 250 nM for WT and up to 1 µM for mutants L27E and R839A, as indicated) in a total volume of 10 µL 1X EMSA buffer (20 mM Tris HCl pH8.0, 100 mM NaCl, 1 mM DTT, 0.1 mg/mL BSA, 5% glycerol) and then incubated for 10 min at 25 ˚C. The samples were then loaded onto a 6% polyacrylamide (29:1) native 1xTBE gel and separated by electrophoresis in 1xTBE at 150V for 40 mins. The gels were visualised using a Typhoon scanner and analysed using ImageQuant software.

## Fluorescence anisotropy

Fluorescence anisotropy measurements were made at 25 ˚C on a Horiba Jobin Yvon FluoroMax fluorimeter. The HEX dye was excited at wavelength 530 nm (excitation slit width 5 nm), and the emission was detected at 550 nm (emission slit width 5 nm). For the direct titrations, a 5 nM solution of DNA substrate in binding buffer (20 mM Tris–HCl (pH 8.0), 20 mM NaCl, 1 mM DTT) was used to which was added increasing amounts of concentrated CtIP protein. Note that the stated CtIP concentrations are as a tetramer. The contents of the cuvette were mixed thoroughly with a pipette and left for 1 min, which was sufficient to reach equilibrium, before each anisotropy reading was taken in duplicate and the numbers averaged. Fluorescence intensity changes were negligible (<10% at saturation). The titrations were carried out three times independently to generate the error bars shown. The competition assays were carried out in the same way but with the pre-formation of the CtIP-Forked DNA complex at 5 nM DNA: 50 nM CtIP$_4$ (yielding about 75% bound DNA in the absence of competitor) followed by the addition of increasing amounts of unlabelled competitor DNA.

Anisotropy is given by *Equation 1*:

$$A = \frac{(I_{VV} - I_{VH})}{(I_{VV} + 2I_{VH})} \tag{1}$$

where $I_{VV}$ and $I_{VH}$ are the intensities of the vertical and horizontal components of the emitted light using vertical polarised excitation. Anisotropy values were normalised (to percent of maximum change) between repeats to account for small differences in the initial and final absolute readings according to *Equation 2*:

$$A(\%) = 100\left(1 - \frac{(A\max - A)}{(A\max - A\min)}\right)$$ (2)

The error bars shown are calculated from normalised values and represent the standard error of the mean calculated from three independent titrations. For the direct titrations of labelled DNA with CtIP, the binding isotherms were generated by plotting protein concentration against normalised anisotropy and were fit using GraphPad Prism to *Equation 3*:

$$B_{obs} = \left(\frac{B_{max} \times [CtIP]}{K_d + [CtIP]}\right) + C$$ (3)

where $B_{obs}$ is the observed binding, $B_{max}$ is the maximal binding signal at saturation, C is an offset value and $K_d$ is the equilibrium dissociation constant.

For competition assays, the binding data were normalised and then fit to a hyperbolic decay function to obtain an $IC_{50}$ value, with which to compare the efficacy of different competitors using *Equation 4*:

$$B_{obs} = \left(\frac{C \times [DNA_{comp}]}{IC_{50} + [DNA_{comp}]}\right) + B_{ini}$$ (4)

Where $[DNA_{comp}]$ is the concentration of the competitor unlabeled DNA (the dependent variable), $B_{ini}$ is the normalized starting anisotropy, C is a scaling value equivalent to total loss of binding signal, and $IC_{50}$ is the total concentration of competitor DNA required to half maximally inhibit binding of the labelled DNA by CtIP. Note that, under the conditions used here, the relationship between $IC_{50}$ and $K_{comp}$ (the equilibrium dissociation constant for CtIP interaction with the unlabeled competitor) is complex (*Huang, 2003*). For weak binding competitors (relative to the probe DNA), the measured $IC_{50}$ value is proportional to $K_{comp}$ whereas for tight binding inhibitors the measured $IC_{50}$ is limited by the affinity of CtIP for the probe DNA.

For experiments with the dephosphorylated CtIP protein where the interaction with DNA was under tight binding conditions, the stoichiometry and affinity of interaction between CtIP and DNA was determined using *Equation 5*:

$$B_{obs} = \left(B_{max} \times \frac{\left(([CtIP] \times K_d \times n[DNA]) - \sqrt{([CtIP] \times K_d \times n[DNA])^2 - (4 \times [CtIP] \times n[DNA])}\right)}{2n[DNA]}\right) + B_{ini}$$ (5)

where $B_{obs}$ is the observed binding (fluorescence anisotropy), $B_{ini}$ is the starting anisotropy, $B_{max}$ is the maximal anisotropy signal at saturation, $K_d$ is the equilibrium dissociation constant and n is the number of CtIP tetramers binding to each DNA molecule. In these fits, the $K_d$ value was held constant at the measured value of 1.5 nM, and the value of n was shared in a global fit of all three data sets collected at different values of [DNA].

## Nuclease assays

Nuclease assays were carried out in 10 μL volumes and were assembled on ice. $CtIP_4$ was titrated against 5 nM Cy5 labelled DNA substrate at 16, 31, 63, 125 and 250 nM in 10 mM Tris-HCl pH 8.0, 1 mM DTT, 250 μg/mL BSA and 0.5 mM $MnCl_2$. Reactions were incubated at 37°C for 120 min and were stopped by the addition of 10 μL 4 mg/mL proteinase K and 1% w/v SDS. Reaction products were separated on 16% w/v acrylamide TBE gels at 200 V for 45 min. Gels were imaged using a Typhoon FLA 9500 scanner (GE Healthcare Life Sciences) and were quantified using ImageQuantLE software (GE Healthcare Life Sciences). To assess divalent cation preference, 250 nM $CtIP_4$ was incubated with 5 nM Cy5 labelled DNA substrate in 10 mM Tris-HCl pH 8.0, 1 mM TCEP, 250 μg/mL BSA and 0.5 mM divalent cation (as indicated) and were then treated as before.

## Acknowledgements

OJW, SJN and MSD were supported by the Wellcome Trust (100401/Z/12/Z). HJK and DBW were supported by Cancer Research UK (C6913/A21608). FM-H acknowledges support from the Spanish Ministry of Science Innovation and Universities (project BFU2017-83794-P, (AEI/FEDER, UE)) and from European Research Council (ERC) under the European Union Horizon 2020 research and innovation (grant agreement No 681299). AM-G acknowledges support from the Spanish MINECO through a PhD fellowship (BES-2015–071244). We are grateful to Kate Heesom, Kelly Sanders, Roz Williamson, Fruszina Rabi and Yuriy Chaban for technical assistance and to Charles Grummit for the gift of human Ku protein.

## Additional information

### Funding

| Funder | Grant reference number | Author |
| --- | --- | --- |
| Wellcome | 100401/Z/12/Z | Oliver J Wilkinson<br>Sarah J Northall<br>Mark Simon Dillingham |
| Cancer Research UK | C6913/A21608 | Haejoo Kang<br>Dale B Wigley |
| Spanish Ministry of Science and Technology | BFU2017-83794-P | Fernando Moreno-Herrero |
| European Research Council | 681299 | Fernando Moreno-Herrero |
| Ministry of Economy and Competitiveness | BES-2015-071244 | Alejandro Martín-González |

The funders had no role in study design, data collection and interpretation, or the decision to submit the work for publication.

### Author contributions

Oliver J Wilkinson, Conceptualization, Formal analysis, Investigation, Methodology, Writing—original draft, Writing—review and editing; Alejandro Martín-González, Haejoo Kang, Sarah J Northall, Formal analysis, Investigation, Methodology, Writing—review and editing; Dale B Wigley, Conceptualization, Formal analysis, Funding acquisition, Writing—review and editing; Fernando Moreno-Herrero, Conceptualization, Formal analysis, Funding acquisition, Investigation, Methodology, Writing—review and editing; Mark Simon Dillingham, Conceptualization, Formal analysis, Funding acquisition, Investigation, Methodology, Writing—original draft, Writing—review and editing

### Author ORCIDs

Dale B Wigley (iD) http://orcid.org/0000-0002-0786-6726
Fernando Moreno-Herrero (iD) http://orcid.org/0000-0003-4083-1709
Mark Simon Dillingham (iD) http://orcid.org/0000-0002-4612-7141

### Decision letter and Author response

Decision letter https://doi.org/10.7554/eLife.42129.026
Author response https://doi.org/10.7554/eLife.42129.027

## Additional files

### Supplementary files

• Supplementary file 1. Comprehensive data for DNA competition assay and details of substrate construction. Supplementary Table 1: $IC_{50}$ values for competitor DNA molecules used in DNA unbinding assays. The reported error is the error associated with the fit to a hyperbolic unbinding curve as shown in *Figure 5—figure supplement 2*. Supplementary Table 2: Assembly/Source of DNA substrates. Small DNA substrates were prepared by annealing different combinations of short

oligonucleotides (A-T). The sequences for the oligonucleotides are presented in Supplementary Tables 3. Supplementary Table 3: Sequences of oligonucleotides used to assemble competitor DNA molecules.

DOI: https://doi.org/10.7554/eLife.42129.022

• Transparent reporting form

DOI: https://doi.org/10.7554/eLife.42129.023

## Data availability

All data generated or analysed during this study are included in the manuscript and supporting files.

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
