## [Decision Letter]

Thank you for submitting your article "CtIP forms a tetrameric dumbbell-shaped particle which bridges complex DNA end structures for double-strand break repair" for consideration by *eLife*. Your article has been reviewed by three peer reviewers, including Maria Spies as the Reviewing Editor and Reviewer #1, and the evaluation has been overseen by John Kuriyan as the Senior Editor. The following individuals involved in review of your submission have agreed to reveal their identity: Luca Pellegrini (Reviewer #2); Claire Wyman (Reviewer #3).

The reviewers have discussed the reviews with one another and the Reviewing Editor has drafted this decision to help you prepare a revised submission.

Summary:

This is a biochemical and biophysical study of CtIP, a DNA repair factor with an important role in DNA-end resection. Activation of human CtIP protein promotes homologous recombination as it cooperates with the DSB processing machinery and is especially important for preparation of complex DSBs for longer range resection. The authors purified the full-length human CtIP. SEC-MALS analysis suggested that the protein is a tetramer, but with an unusual shape. The low resolution EM showed that this unusual shape is a dumbbell, in which polar globular domains are separated by the rod-like structure, which includes the tetramerization core. The dimers of CtIP appear as "tadpoles". Similar, but more flexible structures were obtained by AFM. AFM, however, showed a broader diversity of the structures available to the CtIP tetramer, with most intriguing being the "splayed dimers". Expertly performed biochemical studies show that human CtIP binds DNA structures with complex ends (forks, protein blocks etc.) with high affinity, that dephosphorylation enhances the binding. They also show that CtIP can bind simultaneously two DNA molecules and visualise DNA bridging using AFM. Based on these findings they propose a slide-and-capture model of CtIP function on DNA.

Overall, the reviewers agreed that this is an important, well-executed study that provides significant experimental support for current models of CtIP function. Much of the data solidly supports the authors' conclusions and their elegant "slide-and-capture" model. However, there are a couple of major issues that need to be addressed before publication.

Essential revisions:

1) First, the reviewers were concerned with the AFM imaging data for DNA bridging. The authors need to provide additional explanation or experiments: CtIP has here low nM affinity for DNA end structures used. Binding reactions for AFM had 10 nM DNA and up to 250 nM CtIP BUT still show very little DNA bound protein. Please explain. If due to differences in buffer conditions, repeated with same buffer for all binding reactions.

Additional quantification if the AFM imaging is also necessary: Although the CtIP is ~100kD monomer and 400kD tetramer there is no additional volume/mass evident at the junction of bridged DNAs. What is the actual evidence of protein bound to the DNA in these images? This needs to be clarified. Measurements such as width and height of DNA and the protein coiled-coils could help. Perhaps these are exactly the same and cannot be distinguished, possible though unlikely, can easily be measured from existing images. Never the less the globular parts of the CtIP should still be evident at the junction of bridged DNA, but this is not the case for the examples shown in Figure 6.

Given the absence of protein mass/volume at DNA junctions, the authors need to address (with additional controls or explanation) the possibility that if the substrates were not well purified or separated from DNA with oligo ligated to one end, the DNA may have a sticky end and could form dimer due to hybridization of self-complementary ends.

In the EM analysis, the relative size of the globular domain is rather large, relative to the central connecting rod. In fact, it is surprising that a globular domain should be visible at all, given that the only autonomously folded part of CtIP is its N-terminal region (1-150, ~17% of the protein), while the rest is predicted to be mostly disordered, including the conserved C-terminal DNA-binding domain (~14%). Related to this, the shape of the protein as a dumbbell-like particle suggests that the two chains in each dimer, projecting away from the central, N-tetramerisation motif, manage somehow to remain associated throughout their length. This is an interesting, unanticipated finding of the study and should be highlighted in the Discussion.

2) The second main concern was about the possibility of Streptavidin-CtIP interaction. The positive effect of adding Streptavidin to CtIP's affinity towards DNA is general, which contrasts with the sharply different affinities for the various DNA substrates. Can the authors exclude whether a weak, non-specific interaction between CtIP and DNA-tethered Streptavidin could be responsible for the additional boost in competitor efficacy? After all, the CTD of CtIP is likely to engage in multiple protein-protein interactions. This wouldn't be adequately controlled by placing the streptavidin in the middle of the DNA, because in this substrate the main interaction site would be lost through steric hindrance. Even adding free streptavidin to the sample wouldn't completely control for this, because the effect of a putative non-specific interaction with streptavidin would presumably only be noticeable when taking place concurrently with the main interaction to the DNA. The increased affinity of CtIP for blocked DNA ends is an important conclusion of this study, however, the reviewers agreed that this additional control experiment would be a significant amount of extra-work. Thus, we would leave it to the authors to either carry out this control or to address this specific point in their Discussion.

3) Finally, it would be informative if the authors discuss their vision of how CtIP is targeted to the ends (perhaps through the interaction with MRN and MRN-dependent activation) and not to any other region of the genome that has sufficient stretches of naked DNA between obstacles (e.g. nucleosomes or other DNA bound proteins).

---

## [Author Response]

Essential revisions:1) First, the reviewers were concerned with the AFM imaging data for DNA bridging. The authors need to provide additional explanation or experiments: CtIP has here low nM affinity for DNA end structures used. Binding reactions for AFM had 10 nM DNA and up to 250 nM CtIP but still show very little DNA bound protein. Please explain. If due to differences in buffer conditions, repeated with same buffer for all binding reactions.

For pragmatic reasons our AFM imaging is performed under different conditions compared to the bulk experiments. We tried many methods to reduce the negative charge on the mica surface to facilitate DNA binding and imaging of DNA:protein complexes. We eventually found that treating the surface with moderate MgCl_2_ concentrations (7.5 mM) provided the best conditions. However, bulk experiments also showed that elevated Mg^2+^ concentrations were very poor for imaging CtIP alone and also dramatically decreased the affinity of CtIP for DNA (Author response image 1), and so our final protocol involved a Mg^2+^ “pre-treatment” of the mica, followed by re-introduction of the CtIP:DNA complexes in the standard buffer. Because the pretreatment was necessary for binding, there must be residual Mg^2+^ which is required for immobilisation of the DNA, but which would also be expected to inhibit the CtIP interaction concomitant with deposition. It is of course difficult to define these conditions precisely. Finally, the drying of the mica surface before imaging may also contribute to the low observed binding. We have now added statements in the Materials and methods and Results to more clearly explain this (subsection “CtIP bridges DNA in vitro”, first paragraph and subsection “Imaging”). ^-^

**Author response image 1. respfig1:** Moderately elevated MgCl_2_ concentrations dramatically reduce the affinity of CtIP for DNA.

Additional quantification if the AFM imaging is also necessary: Although the CtIP is ~100kD monomer and 400kD tetramer there is no additional volume/mass evident at the junction of bridged DNAs. What is the actual evidence of protein bound to the DNA in these images? This needs to be clarified. Measurements such as width and height of DNA and the protein coiled-coils could help. Perhaps these are exactly the same and cannot be distinguished, possible though unlikely, can easily be measured from existing images. Never the less the globular parts of the CtIP should still be evident at the junction of bridged DNA, but this is not the case for the examples shown in Figure 6.

We have performed additional quantification of our AFM images and have added this information to the Results (subsection “CtIP bridges DNA in vitro”, first paragraph, and Figure 6—figure supplement 1). We do in fact see examples of DNA bridges with additional volume present at the junction which indicates CtIP binding, and these are now presented along with discussion in the legend. Nevertheless, the reviewers are quite right to point out that we also see examples in which there is no apparent CtIP at the junctions and we imagine that these could arise in two ways. Firstly, for the reasons elaborated above, CtIP may dissociate from DNA concomitant with deposition onto the mica surface. Secondly, we do expect a small fraction (~4%) of the apparent bridges between DNA to be protein-free based on our own control experiments, which show longer than single contour length in the absence of protein (Figure 6C). This “noise” increases in the presence of proteins that cannot bind DNA (e.g. BSA) presumably due to crowding effects on the surface of the mica.

Given the absence of protein mass/volume at DNA junctions, the authors need to address (with additional controls or explanation) the possibility that if the substrates were not well purified or separated from DNA with oligo ligated to one end, the DNA may have a sticky end and could form dimer due to hybridization of self-complementary ends.

We think not for two reasons. (1) The bridging substrates were prepared by the annealing of 150-fold excess of oligonucleotides containing poly-dT overhangs to minimise this kind of issue. They were subsequently gel-purified away from unligated DNA which had a different mobility in agarose. Nevertheless, there might be some trace unligated DNA left, especially with just a single unligated sticky end which may not resolve well from the desired product. (2) Regardless of the substrate quality, the elevated levels of bridging we observe in the presence of CtIP cannot arise because of DNA:DNA interactions. Control experiments (bridging measured in the absence of protein, in the presence of BSA, or in the presence of CtIP mutants that do not bind DNA) all show markedly less bridging than experiments with wild type CtIP (or CtIP λ) and all were performed with identical DNA molecules. Moreover, the observed bridging is sometimes “end-to-centre” or “centre-to-centre” which is presumably impossible if it arises from sticky end annealing. We have now clarified these points in the Materials and methods (subsection “Sample preparation”) and Results (subsection “CtIP bridges DNA in vitro”, first paragraph) sections.

In the EM analysis, the relative size of the globular domain is rather large, relative to the central connecting rod. In fact, it is surprising that a globular domain should be visible at all, given that the only autonomously folded part of CtIP is its N-terminal region (1-150, ~17% of the protein), while the rest is predicted to be mostly disordered, including the conserved C-terminal DNA-binding domain (~14%). Related to this, the shape of the protein as a dumbbell-like particle suggests that the two chains in each dimer, projecting away from the central, N-tetramerisation motif, manage somehow to remain associated throughout their length. This is an interesting, unanticipated finding of the study and should be highlighted in the Discussion.

We partly agree with the reviewers’ points and have modified the Discussion accordingly (first paragraph).The relative size of the globular domains is approximately as expected given that they account for over 80% of the protein, but it is perhaps surprising that they appear to be (at least partially) ordered. It should however be remembered that the EM “tadpoles” could conceivably represent tetramers with unfolded termini at one end and also that the particles we observe in AFM are far more variable, and include examples of tetrameric dumbbells with both “splayed” as well as intact globular C-terminal ends. The idea that the C-terminal ends are associated is also consistent with the fact that CtIP binds 0.5 DNA molecules per monomer, and implies that the DNA binding site might be dimeric in nature.

2) The second main concern was about the possibility of Streptavidin-CtIP interaction. The positive effect of adding Streptavidin to CtIP's affinity towards DNA is general, which contrasts with the sharply different affinities for the various DNA substrates. Can the authors exclude whether a weak, non-specific interaction between CtIP and DNA-tethered Streptavidin could be responsible for the additional boost in competitor efficacy? After all, the CTD of CtIP is likely to engage in multiple protein-protein interactions. This wouldn't be adequately controlled by placing the streptavidin in the middle of the DNA, because in this substrate the main interaction site would be lost through steric hindrance. Even adding free streptavidin to the sample wouldn't completely control for this, because the effect of a putative non-specific interaction with streptavidin would presumably only be noticeable when taking place concurrently with the main interaction to the DNA. The increased affinity of CtIP for blocked DNA ends is an important conclusion of this study, however, the reviewers agreed that this additional control experiment would be a significant amount of extra-work. Thus, we would leave it to the authors to either carry out this control or to address this specific point in their Discussion.

In our anisotropy-based DNA binding competition experiments we used streptavidin:biotin moieties as models for nucleoprotein conjugates at DNA ends. These have been widely used in the related literature to mimic the more physiologically relevant blocks that are formed by (e.g.) Spo11, etoposide-poisoned TopoII, or tightly bound NHEJ factors such as Ku. For example, the conclusion that CtIP activates the intrinsic nuclease activity of MRN at blocked DNA ends arose from analogous experiments to these, and so our experiments are important in this context. Nevertheless, the reviewers are right to point out that it is difficult to completely exclude the possibility that a weak non-specific interaction could also explain the effect we observe in our assay, and we now acknowledge this in the text (subsection “CtIP binds preferentially to DNA substrates with blocked DNA ends”, third paragraph). However, we are not clear what additional control could be performed, or is being suggested, since we have already performed the two that are mentioned in the review (moving the position of the streptavidin to the middle of the substrate and adding free streptavidin instead of DNA-conjugated streptavidin). We envision the best experiments moving forward will use more physiologically relevant covalent blocks; we are setting up to investigate the interaction of CtIP with topoII-conjugated DNA for instance (note that one cannot use non-covalent blocks in these competition experiments as they will exchange with the probe DNA to interfere with the FA measurement).

In addition to the above, we have performed a new experiment (subsection “CtIP binds preferentially to DNA substrates with blocked DNA ends”, last paragraph, Figure 5—figure supplement 4) which, we hope the reviewers will agree, helps to better define and highlight the unusual DNA binding properties of CtIP. One of our major conclusions supported by the streptavidin FA experiments is that CtIP is not a DNA end binding protein per se, but that modifications to ends affect its ability to retain DNA, which we rationalised in terms of a sliding model. This is in complete contrast to “classical” DNA end recognition factors such as Ku, or bacterial resection complexes like RecBCD, which display structural specificity for free DNA ends. We now present equivalent competition FA experiments with Ku which show that streptavidin blocks severely inhibit the ability of linear DNA substrates to compete for binding (behaviour which is the complete opposite to that of CtIP).

3) Finally, it would be informative if the authors discuss their vision of how CtIP is targeted to the ends (perhaps through the interaction with MRN and MRN-dependent activation) and not to any other region of the genome that has sufficient stretches of naked DNA between obstacles (e.g. nucleosomes or other DNA bound proteins).

We accept this point and have added a short paragraph to the Discussion (fourth paragraph).Briefly, and as implied by the reviewers, the most obvious possibility is that CtIP is directed to DNA ends by virtue of its many interactions with partner proteins. These include the BRCA1-BARD1 complex and MRN, either of which might influence its binding location via assembly or by facilitating post-translational modification.